# Applications of Exosomes in Diagnosing Muscle Invasive Bladder Cancer

**DOI:** 10.3390/pharmaceutics14102027

**Published:** 2022-09-23

**Authors:** Jillian Marie Walker, Padraic O’Malley, Mei He

**Affiliations:** 1Department of Pharmaceutics, College of Pharmacy, University of Florida, Gainesville, FL 32611, USA; 2Department of Urology, College of Medicine, University of Florida, Gainesville, FL 32611, USA

**Keywords:** muscle invasive bladder cancer, exosomes, biomarkers, bladder cancer screening, bladder cancer diagnosis

## Abstract

Muscle Invasive Bladder Cancer (MIBC) is a subset of bladder cancer with a significant risk for metastases and death. It accounts for nearly 25% of bladder cancer diagnoses. A diagnostic work-up for MIBC is inclusive of urologic evaluation, radiographic imaging with a CT scan, urinalysis, and cystoscopy. These evaluations, especially cystoscopy, are invasive and carry the risk of secondary health concerns. Non-invasive diagnostics such as urine cytology are an attractive alternative currently being investigated to mitigate the requirement for cystoscopy. A pitfall in urine cytology is the lack of available options with high reliability, specificity, and sensitivity to malignant bladder cells. Exosomes are a novel biomarker source which could resolve some of the concerns with urine cytology, due to the high specificity as the surrogates of tumor cells. This review serves to define muscle invasive bladder cancer, current urine cytology methods, the role of exosomes in MIBC, and exosomes application as a diagnostic tool in MIBC. Urinary exosomes as the specific populations of extracellular vesicles could provide additional biomarkers with specificity and sensitivity to bladder malignancies, which are a consistent source of cellular information to direct clinicians for developing treatment strategies. Given its strong presence and differentiation ability between normal and cancerous cells, exosome-based urine cytology is highly promising in providing a perspective of a patient’s bladder cancer.

## 1. Introduction

The bladder is part of the urinary system, located in the pelvic cavity, and serves as a short-term reservoir for urine [1,2,3]. The bladder is comprised of four layers as depicted in Figure 1. The innermost layer is the transitional epithelium [4]. This tissue type is important in bladder structure due to flexible stretching ability. Transitional epithelium’s high elasticity allows for substantial surface area to store urine. The next layer out is the lamina propria which serves to reinforce the inner lining and is made of elastic connective tissue. The third layer of the bladder is the smooth muscle layer also known as the detrusor muscle, which is three layers thick and contracts to evacuate urine. The fourth outermost layer of the bladder is the serosal layer, which serves as a barrier to decrease friction between the bladder and surrounding organs.

Bladder cancer manifests as two broad classifications, non-muscle invasive bladder cancer (NMIBC) and muscle invasive bladder cancer (MIBC). The pathophysiology of bladder cancer is described as a dual pathway and can be best defined by the presence of papillary and nonpapillary lesions [3,4]. NMIBC accounts for 75% of all bladder cancer cases, it does not extend beyond the first layer of the bladder [5]. Papillary lesions typically indicate non-muscle invasive bladder cancer (NMIBC), which accounts for most bladder cancer diagnose. NMIBC is characterized by chromosome nine deletion, which holds the CDKN2A gene that codes for tumor suppressor proteins [6]. Additionally, mutations arise on fibroblast growth factor receptor 3 (FGFR3), PI3K, and telomerase reverse transcriptase (TERT) [7,8,9,10]. Muscle invasive bladder cancer is the second route of the dual pathway as illustrated in Figure 1. It is defined by tumor infiltration beyond the epithelial layers of the bladder and into the detrusor muscle. MIBC also has a deletion of chromosome 9 and mutations in FGFR3, PI3K, TERT [11,12,13]. Additionally, retinoblastoma protein 1 (Rb1) and p53 are mutated and/or deleted [14,15,16,17,18]. In MIBC, Rb1 is truncated which promotes tumorigenesis. There is also an ~50% increase in p53 mutations in muscle invasive bladder cancer which leads to impaired DNA repair capabilities and loss of function in p53 associated tumor suppressor genes [17]. MIBC accounts for 25% of all bladder cancers and, depending on the aggravating factors, has a five-year survival of 70% [19]. The dual pathway of this malignancy contributes to the intricacies in presentation, diagnosis, and treatment.

In 2022, there will be approximately 81,000 new cases of bladder cancer in America [19]. Of those cases, 25% will be MIBC. Muscle invasive bladder cancer is more predominant in men than women with a median age at diagnosis of 73. It is also twice as common in White men than Black or Latino men. Modifiable risk factors for muscle invasive bladder cancer include environmental exposures to aromatic amines, cigarette smoke, and chronic bladder infections. Non-modifiable risk factors for muscle invasive bladder cancer include a family history of bladder cancer and being diagnosed with Lynch syndrome [20].

MIBC often presents with a myriad of symptoms such as hematuria, dysuria, and general constitutional symptoms. These symptoms may be transient, and it should be noted that it can be observed in other non-malignant urogenital disease states. Due to the ambiguity of bladder cancer presentative symptoms, delayed diagnosis is often among patients. The later a patient is diagnosed, the higher the risk for initial diagnosis with MIBC. Patients presenting with concerning symptoms will undergo a comprehensive clinical work-up. A work-up may include a urologic evaluation, radiographic imaging with a CT scan, urinalysis, and cystoscopy. Of the available diagnostic methods, cystoscopy is currently the standard for bladder cancer diagnosis [21,22,23]. Once a diagnosis is confirmed, bladder cancer will be staged using the TNM staging system. Bladder cancer is considered MIBC once it is determined a T2 lesion is present. A T2 lesion or higher indicates the invasion of the cancer into the muscular layer of the bladder. The associated risk for metastasis is higher with muscle invasive bladder cancer and metastatic sites can be local or distant. Local metastatic sites include adipose tissue, lymph nodes, and the peritoneum. Distant sites include the patient’s bones, liver, and lungs [24]. The gold standard of care treatment for muscle invasive bladder cancer is cisplatin-based neoadjuvant chemotherapy with radical cystectomy.

## 2. Biomarkers in Bladder Cancer Diagnosis

Liquid biopsy is popular in cancer diagnostics because samples are collected with less invasive means than solid tissue biopsy. Liquid biopsy can be employed for diagnostics, prognosis, and theragnostic [22,23,25,26,27]. There are several types of samples which can be used for biopsy, including blood, urine, and cerebrospinal fluid (CSF) [21,28]. Biomarkers are biological molecules found within a given biopsy specimen used to detect and monitor illnesses and/or conditions. Popular biomarkers of interest in cancer include circulating free DNA (cfDNA), circulating tumor cells (CTAs), circulating proteins and cytokines, circulating extracellular vesicles and exosomes, and T-cells [29,30]. The generation and unique clinical utility of exosomes are the focus of this review. Of the types of liquid biopsy, urine samples are optimal for application in bladder cancer, because urine is stored in the bladder and has the most direct source connecting with bladder cancerous cells [31,32]. There are several urinary biomarker tests approved by the FDA which will be discussed in this review. However, none of them can achieve all the qualities required for a clinically useful urinary biomarker test for bladder cancer diagnosis. An ideal urinary biomarker test needs to incorporate several components including specificity, sensitivity, cost-effectiveness, and ease of interpretation [26,27,33]. The low false-positive rate and low risk of false negatives and undiagnosed disease progression are essential. The cost-effectiveness is critical for both health systems to employ on a large scale and for patients to pay for.

## 3. Exosomes in Muscle Invasive Bladder Cancer

### 3.1. Exosomes Defined

Exosomes are a subgroup of extracellular vesicles (EVs) in size range < 200nm, and derived from the membranes of multivesicular bodies [34,35]. They are released from several cell types, including diseased, malignant, and normal cells. Across the cell types, exosomes communicate and influence physiology at local and distant sites within the body. Exosomes are enriched with CD63, and can be found in blood, breast milk, urine, serum, saliva, mesenchymal, tumor, and dendritic cell samples [36,37,38,39]. They can also cross several internal barriers such as the blood-brain-barrier, the retinal barrier, stromal barrier, placental barrier, and cerebral spinal fluid barrier [40,41,42,43]. Exosomes are heterogenous by their surface molecules and cargos such as proteins, lipids, mRNA, miRNA, lncRNA, and DNA [44]. mRNA is implicated in tumor progression and metastasis through the abnormal upregulation of anion transport, cell growth factors, and the MAPK cascade [45]. Exosome miRNAs have roles in regulation of gene expression and tumor microenvironment in both healthy and malignant cells [46]. lncRNA contributes to the growth and survival of tumors [47]. Exosome DNA may protect tumor cells from regulatory inflammation processes, in turn supporting tumor survival [48]. Exosome biogenesis is largely supported by the endosomal sorting complex required for transport (ESCRT) and micro-vesicular bodies (MVB) [49,50,51]. Prior to exosome release, they are loaded with cargos which may consist of miRNA, proteins, and/or lipids. Once loaded with cargo, further release is regulated via synergistic ESCRT dependent and independent pathways [52]. They are selectively up-taken into cells via endocytosis, receptor–ligand interaction, or cellular membrane fusion [52]. Within the scope of cancer, exosomes are implicated in cancer development and survival [53,54,55,56].

Exosomes have a prominent role in cellular communication, which may lead to the promotion of malignancies as tumors release exosomes carrying pro-tumor genetic information. These pro-tumor exosomes mediated actions are illustrated in Figure 2. Through autocrine interaction, exosomes can change the direction of exosome releasing cells leading to tumor promotion [53,55,57,58,59]. Via paracrine interactions, exosomes can modulate intracellular interaction and the microenvironment of the cells. Angiogenesis is promoted by exosomes, especially in hypoxic conditions [60,61,62], which leads to downstream signaling cascades that can promote malignancies [62]. Cancer promoting histological changes are also influenced by exosomes. They are thought to be highly involved in the epithelial to mesenchymal transition (EMT) malignant lesions undergo as cancer develops [63]. Lastly, exosomes are impactful in angiogenesis to grow and maintain tumor survival. Cells under stress or in hypoxic conditions often release more exosomes [64,65]. Cancerous cells are under stress and experience hypoxic conditions, in turn, an increased release of exosome and signaling is observed [66,67]. Exosomes also contribute to pre-metastatic environments and metastasis. In pre-metastatic environments, exosomes are released from tumors and sent to distant sites to condition the environment into a suitable tumor micro-environment [68,69,70].

### 3.2. Exosomes Involvement in MIBC Progression

The role of exosomes in MIBC progression is not fully understood in current literature. It has been suggested that high quantity of exosomes from MIBC may attribute from the pro-cancer actions such as increased tumor growth, invasion, and angiogenesis [71,72,73]. Starting with proliferation, tumor-derived extracellular vesicles (TEVs) and exosomes alter the operations of tumor suppressor genes to create a protumor microenvironment [74]. The accompanying hypoxia often found in tumor microenvironments further supports the actions of TEVs [75,76,77,78]. A key role of exosomes in high grade tumors and eventual MIBC is promoting metastasis. Exosomes support metastasis by carrying, transferring oncogenic cargoes, and hindering tumor suppressor exosomes [75]. Examples of such activity are evident in bladder cancer exosomes activating the ERK1/2 MAP kinase signaling pathway to promote malignancy of low tumor grade bladder cancer cells [79]. As bladder cancer metastasis continues, the tumor has a higher chance of developing into MIBC. A cornerstone characteristic of muscle invasive bladder cancer is the epithelial to mesenchymal transition (EMT). EMT describes the process of epithelial, urothelial cells in the case of bladder cancer, transforming into mesenchymal tissue (Figure 2C). Mesenchymal tissue can support carcinogenesis which contributes to larger, faster-growing tumors. The clinical significance of the processes from EMT is the development into higher grade aggressive tumors as MIBC. Due to the fast-growing nature of bladder cancer tumors, especially with upregulated pro-tumor EVs and exosomes, MIBC could lead to a complicated clinical picture. 

Exosomes are highly implicated in the development and progression of muscle invasive bladder cancer [75,76]. Several in vitro studies have described the presence of carcinogenic activity being mediated by exosomes. In vitro exosomes demonstrate cellular communication between cancerous bladder cells and histologically diverse tissue, supporting the proposal of exosome mediated metastasis [80,81]. Carcinogenic exosomes had an increase in unfolded endoplasmic reticulum proteins, which leads to the oxidative stress response mechanism within cells when they are proliferating quickly [82]. This oxidative stress phenomenon was observed in mice models utilizing bladder cancer cells [83,84]. Demonstrated using muscle invasive bladder cancer cells, the presence of EMT underscores the impact exosomes have in the setting of MIBC [84]. It was observed that exosomes induce and promote the upregulation of mesenchymal markers in urothelial cells [73,85]. Lastly, exosomes have been documented as upregulating Bcl2 and Cyclin D which promote tumorigenesis [84]. These findings support the role of exosomes in cancer as well as the specific contributions in muscle invasive bladder cancer [82,85,86]. Additionally, to the above findings, there are several exosome biomarkers implicated in bladder cancer tumorigenesis [85,87,88,89]. These urine-based markers are characterized in Table 1. Exosomes established role in muscle invasive bladder cancer and documented urine biomarkers yield opportunity for an exosome-based urine biomarker test for MIBC diagnosis.

### 3.3. Exosome Biomarkers for Muscle Invasive Bladder Cancer Diagnosis

Urinary exosome biomarkers are not currently used as diagnostic tools for MIBC detection. However, exosomes would make an excellent source for biomarkers for several reasons. Exosomes participate in cell-to-cell communication and stimulation of immune responses [90,91]. They can receive feedback and respond to their cellular environment. This quality can be manipulated to identify biomarkers to detect tumors. Additionally, exosomes are reported to be released in a larger quantity in malignant cells than healthy cells [92,93]. So, exosome biomarkers used in a biomarker test would reveal clear results segregating healthy from diseased areas of the bladder. Table 1 lists the biomarkers that are present in urinary exosomes and are implicated in the promotion of bladder cancer. It should be noted that several of these biomarkers are found in other tumor types and therefore not unique to bladder cancer, such as EDIL-3 found also in sarcomas exosomes modulating angiogenesis [94]. The main modes of action for these biomarkers include an increase in tumor cell migration, proliferation of tumor cells, angiogenesis, decreased apoptosis of cancer cells, and pro-tumor microenvironment support. Although this table list is not an exhaustive list of all the discovered biomarkers, it is inclusive of well described urinary exosomes derived biomarkers. Current research suggests there are many more urinary exosome biomarkers to be discovered. 

**Table 1 pharmaceutics-14-02027-t001:** Identified Urinary Biomarkers for Bladder Cancer.

Urinary Biomarkers	EV Source	Mechanism of Action	Effect	Reference
CD36	Urine protein	Increases fatty acid uptake	Increase migration, proliferation, and angiogenesis	[84,95,96]
CD73	Urine protein	Regulates cellular signaling	Increase migration, proliferation, and angiogenesis	[84,95,96]
CD44	Urine protein	Docks proteases on cell membrane	Increase migration, proliferation, and angiogenesis	[84,95,96]
CD9	Urine protein	Exosome mediation of metastasis in conjunction with NUGC-3 and OCUM-12	Promotion of tumor invasion and metastasis	[96,97]
TSG101	Urine protein	regulates ubiquitin-mediated protein degradation, cellular transcription, cell proliferation, and division.	Promotes an increase in downstream cellular stress	[98]
EDIL-3	Urine protein gene	Promotes angiogenesis and metastasis in malignant environments	Enhances the aggressiveness and growth of the tumor/s	[76]
Alpha 1-antitrypsin	Urine protein	Immunity regulation	Decrease apoptosis	[99]
MUC1	Urine protein gene	Promotes histological morphologies and metastasis through several routes of cellular communication	Cancer progression and metastasis	[100]
MUC4	Urine protein gene	Enhances the EMT process and influences immunomodulation	Promotes aggressive metastatic cancers	[101]
MAGE-B4	Urine protein	Increase tumorigenesis and proliferation	Increase ubiquiation and degradation of p53	[89,95]
miR-21	Urine miRNA	Decrease AKT and MAPK pathways	Increase invasion	[77,84,95]
GALNT1	Urine RNAs, lncRNAs	Mediates O-linked glycosylation of sonic hedgehog to promote its activation	Maintains bladder cancer stem cells and bladder tumorigenesis	[95,102,103,104]
UCA1	Urine RNAs, lncRNAs	Regulates CREB	Increase proliferation	[95,102,103,104]
MALAT-1	Urine RNAs, lncRNAs	Antagonize miR-125b	Decrease apoptosis	[95,102,103,104]
UCA 201	Urine RNAs, lncRNAs	Increase the expression levels of ZEB1 and ZEB2 decrease the expression of hsa-miR-145 and the downstreamactin-binding protein FSCN1	Increase migration and invasion	[95,102,103,104]

## 4. FDA Approved Urine-Biomarker Tests for Bladder Cancer

In addition to the standard of care in bladder cancer for diagnosis as cystoscopy, there are several urine based methods for bladder cancer diagnosis, including florescence cystoscopy, urine cytology, urine-based marker detection, and urinary tract imaging as the well-recognized methods to support cystoscopy findings [105]. Urine-based markers are comprised of proteins emitted from tumors, DNA, RNA, exosomes, or other cellular components. They are of particular interest because the collection is inexpensive and non-invasive in terms of developing detection methods with good sensitivity even in low-grade tumors. Currently, six FDA approved urine biomarker tests for bladder cancer have been depicted and summarized in Table 2.

The NMP22BC test kit is a protein-based immunoassay test for bladder cancer diagnosis which utilizes the biomarker nuclear matrix protein 22 (NMP-22). NMP-22 is a cellular protein, which after bladder cell apoptosis, is increased in its release into the urine. Its median sensitivity is ~61% and its specificity is 71% [106]. The NMP-22 BladderChek has an advantage over the NMP22 test because it can be rapidly completed in 30 min. NMP-22 BladderChek is approximately 55.7% specific and approximately 85.7% sensitive at the 95% confidence interval. Both the NMP-22 BladderChek and NMP22BC test kit are affected by tumor grade, staging and concurrent genitourinary pathologies.

The BTA TRAK and BTA stat are quantitative and qualitative tests, respectively. BTA stat is an adjunctive rapid immunochromatographic assay to cystoscopy. It utilizes monoclonal antibodies to identify complement-factor H-related protein, associated with bladder cancer, to identify malignancy. The BTA stat sensitivity is 67% and the specificity is 70% [107], and both values are influenced by the presence of other urinary conditions because it can confound the results of the test. BTA TRAK is not as widely used as its stat counterpart, which may be due to its high false-positive and negative rate. BTA Trak’s median sensitivity is ~75.5% and its median specificity is 53.5%.

ImmunoCyt/uCyt+ is used as an adjunctive test to cystoscopy for monitoring recurrent bladder cancer. It is an immunocytochemical test that utilizes three fluorescent antibodies. The corresponding antigens include two mucins associated with bladder cancer and one carcinoembryonic antigen which are only found in exfoliated cancerous bladder cancer cells. Their sensitivity and specificity are both 78% [96].

Lastly, UroVysion utilizes fluorescence in situ hybridization (FISH) to detect bladder cancer. It has a clinical sensitivity of 75% and specificity of 93% [108]. It should be noted that this assay has a profound anticipatory effect due to its sensitivity. Thus, it is imperative that positive test using this method are closely monitored. There are several options available for cytological bladder cancer detection. However, as described with each available test, further improvement regarding sensitivity and specificity of the measurements is needed. Table 2 listed the six FDA approved urinary biomarker tests for bladder cancer. None of these tests are used alone for bladder cancer diagnosis due to their low sensitivity and specificity. The ideal urinary biomarker tests for bladder cancer would need to have high specificity, high sensitivity, being cost-effective, and easy to replicate.

**Table 2 pharmaceutics-14-02027-t002:** FDA Approved Urine Biomarker Tests for Bladder Cancer.

Test	Type of Test	Biomarker Tested	Sensitivity	Specificity	Reference
NMP22 BC test kit	Sandwich ELISA	NMP22	61%	71%	[106]
NMP22 Bladder Check	Sandwich ELISA	NMP22	55.7%	85.7%	[106,109]
BTA TRAK	ELISA	Complement factor H-related protein	75.5%	53.5%	[106]
BTA stat	Sandwich ELISA	Complement factor H-related protein	67%	70%	[106,110]
ImmunoCyt/uCyt	Immunofluorescent cytology	Monoclonal antibodies	78%	78%	[106,111,112]
UroVysion	FISH	DNA of malignant urothelial cells	75%	93%	[106,113,114,115]

## 5. Muscle Invasive Bladder Cancer Diagnostic Tests in Clinical Trials

Presently, only two exosome-based biomarker diagnosis for muscle invasive bladder cancer clinical trials currently being conducted as summarized in Table 3. However, a few more clinical trials are investigating other circulating biomarkers including cell free DNAs in muscle invasive bladder cancer. An American multi-facility observational cohort study called Clinical Performance Evaluation of the C2i-Test, MIBC patients are submitting blood samples for detection of molecular residual disease via ctDNA analysis [116]. The measured primary outcome is predicting three-year recurrence free survival post definitive treatment. The AURORAX-0093A: Glycosaminoglycan Profiling for Prognostication of Muscle-invasive Bladder Cancer—a Pilot Study (AUR93A) is an observational cohort study based in Italy and Sweden, which is utilizing glycosaminoglycan profiling scores to determine the prognosis of MIBC. The primary outcome is the proportion of patients who have complete response at the first post-radical cystectomy visit. The Samsung Medical Center in Seoul, Republic of Korea is conducting an observational cohort study called Clinical Utility of VI-RADS in Diagnosis of MIBC, which is studying the application of Vesical Imaging Report and Data System (VI-RADS) in Diagnosis of Muscle Invasive Bladder Cancer. The primary endpoint is measuring the accuracy of the VI-RADS scoring system in MIBC diagnosis [117]. Currently, there are 92 in total as the pioneer clinical trials utilizing exosomes for diagnosing cancers mainly including lung cancer, breast cancer, pancreatic cancer, prostate cancer, and colorectal cancer. The absence of clinical trials involving exosomes in MIBC diagnosis indicates the need for more research in this area.

## 6. Discussion

The bladder is a urinary reservoir consisting of four distinct layers: (1) the transitional epithelium; (2) lamina propria; (3) detrusor muscle; (4) and serosal layer. Bladder cancer can develop into non-muscle invasive bladder cancer or muscle invasive bladder cancer. These tumor growth patterns are best described in the dual development pathway of bladder cancer. Papillary lesions have mutations and cellular disorders, however, there is no infiltration to the bladder tissue beyond epithelium. Non-papillary lesions have further disorder due to deletion of p53 and Rb1. These lesions extend to the detrusor muscle and are considered a higher risk version of bladder cancer. Bladder cancer is highly prevalent in the United States and 25% of all bladder cancer cases will be muscle invasive bladder cancer. This cancer mainly affects elderly white men. Modifiable risk factors for developing ladder cancer include environmental exposures to aromatic amines, cigarette smoke, and chronic bladder infections. To diagnose bladder cancer, patients will undergo a work-up inclusive of a urologic evaluation, radiographic imaging with a CT scan, urinalysis, and cystoscopy. Many efforts have been made to replace this procedure with a less invasive method of diagnosis, such as urine biomarker tests. However, due to the low sensitivity and specificity of currently available tests, urinary biomarkers have not been able to replace cystoscopy in bladder cancer diagnosis. Urinary exosomes are the promising alternative. Exosomes are a key biological player in the development of MIBC and implicate several pro-tumor actions such as tumor proliferation, metastasis, and survival. Exosome’s largest role in muscle invasive bladder cancer is the epithelial to mesenchymal transition, which has been observed that exosomes induce and promote the upregulation of mesenchymal markers in urothelial cells [118]. This transition leads to a strong pro-tumor microenvironment within the layers of the bladder. EMT supported tumor growth, which can lead to more invasive bladder cancer such as muscle invasive bladder cancer. Identifying exosome biomarkers strongly associated with the EMT process, which will progress the strategies employed to diagnose invasive bladder cancers early and save patients’ lives. 

EVs and Exosomes are a promising source of biomarkers for muscle invasive bladder cancer. However, they are not without flaws. Exosomes have pitfalls in their isolation and purification methods. The Minimal information for studies of extracellular vesicles 2018 ([119,120]) describes several isolation and purification techniques [121]. The consensus is ultracentrifugation and ultrafiltration may have the largest amount of yield, however, it will co-isolate other membrane particles and protein aggregate. As described by Doyle and Wang, these two methods have a high risk of destroying exosomes in the process of isolation and purification leading to low yield amounts of pure exosomes [91]. From a clinical standpoint, it is imperative that standards be established for exosome biomarker characteristics to accommodate for the variety of patient populations [122] and to accommodate for the natural heterogeneity of patient populations. Therefore, the isolation and purification methods for extracting exosomes from various clinical fluids are critical. The EVs and exosomes isolation techniques have been intensively developed in the past decade. Vast amount of review papers regarding EV isolation techniques have been reported [123,124,125,126,127,128,129,130,131,132,133,134,135,136]. The well-documented methods for isolating exosomes from biological samples include, but not limited to, differential ultracentrifugation, size-exclusion chromatography and immunoaffinity capture [137]. The MISEV describes the application of the exosomes as the deciding factor in the type of separation to use [121]. The downstream characterization of EV and exosome quality and biomarker expression is also challenging. The well accepted nanoparticle tracking analysis suffers from largely scattered variations [138]. In the case of clinical research and for application in biomarker identification, collecting the purest EV population is imperative. It is recommended to employ ultracentrifugation in conjunction with ultrafiltration as a conventional approach, due to their wide accessibility and cost-effectiveness compared to other isolation methods. However, ultracentrifugation has a low recovery rate of 2–25%. Note that recovery and purity of exosomes are dependent on the density, size, quantity, and molecular relevance of the sample [91]. Ultrafiltration is subject to EV destruction due to the shear force of membrane filtration. Currently, affinity purification such as immunomagnetic beads and affinity column are getting more recognition in terms of homogeneous population and purity relevant to interests [139]. Due to variance in EV isolation and variable return of results from widely accepted methods, it is important that highly specific and sensitive isolation techniques are developed in the future for improving diagnostic outcomes. 

The tests currently approved by the Food and Drug Administration (FDA) for bladder cancer diagnosis do not unlock the power of exosomes. Given the vast array of exosome biomarkers identified in the development and survival of bladder cancer, the huge needs of exosome biomarker diagnostic test are presented. For developing exosome based diagnostic test, the well-established exosome isolation and characterization are critical and need to be standardized. Some pioneer research work has been reported recently to overcome the isolation challenge and ensure the exosome purity and specificity [84,96,139]. However, clinical translation is still lacking. More efforts on clinical translation will be needed in the future research.

## Figures and Tables

**Figure 1 pharmaceutics-14-02027-f001:**
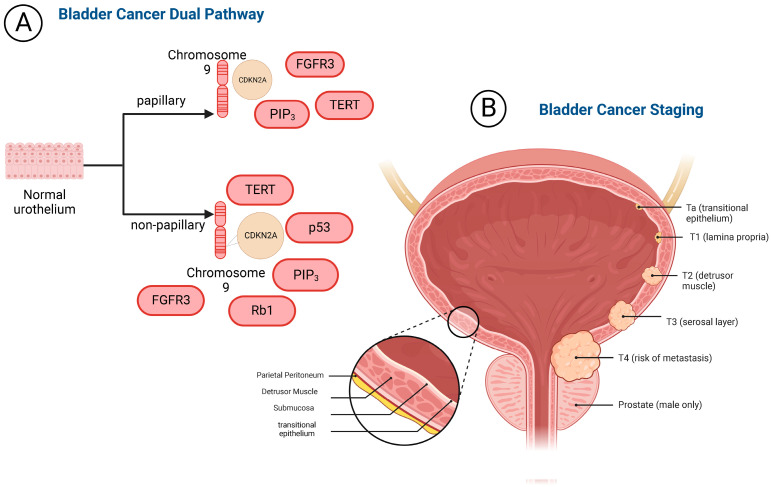
(**A**) The bladder cancer dual pathway involves the presence of papillary or non-papillary lesions. papillary lesions harbor mutations in FGFR3, TERT, PIP3, and deletion of CDKN2A on Chromosome 9. Papillary lesions typically present as NMIBC. Non-papillary lesions include mutation of TERT, PIP3, FGFR3, p53, Rb1, and deletion of CDKN2A on chromosome 9. Non-papillary lesions typically describe MIBC. (**B**) The bladder consists of four layers: the epithelium, submucosa, detrusor muscle, and parietal peritoneum. Staging for bladder cancer is depicted in this figure. Tumors are classified based on the TNM grading system where T describes the primary tumor in terms of its size and tissue penetration. N characterizes the involvement, or lack thereof, lymph nodes. M describes the presence of absence of metastasis. MIBC is characterized by being T2 and can present with or without nodal or metastatic involvement.

**Figure 2 pharmaceutics-14-02027-f002:**
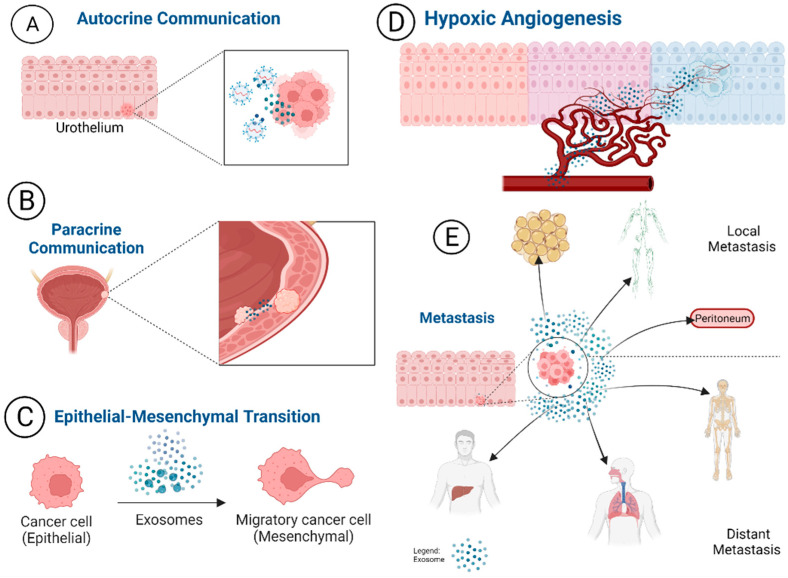
Exosomes are involved in cancer cell development, proliferation, and survival. (**A**) Through autocrine communication, exosomes communicate to the cells they are released from to promote a suitable microenvironment for tumors and contribute to the activation of pro-tumor mutations within their host cell. (**B**) Paracrine communication allows exosomes to communicate to nearby cells. They can modify the signaling pathways leading to changes in gene expression of surrounding cells. (**C**) Exosome’s role in the epithelial to mesenchymal transition (EMT). (**D**) In hypoxic conditions exosomes can promote the growth of new blood vessels, often called angiogenesis. Through this process, nutrients can be sent to malformed cells to support their growth and proliferation into cancer cells. (**E**) Through exosome regulation of cellular communication, changing the histology of the tissue could create a more suitable tumor microenvironment. This process supports more malignant forms of cancer. The cellular communication of exosomes to distant sites allows them to prepare distant organs for later infiltration of tum or cells.

**Table 3 pharmaceutics-14-02027-t003:** Clinical Trials Utilizing Extracellular Vesicles and Exosomes as Biomarkers for Bladder Cancer Diagnosis. Information is from searching via clinicaltrials.gov.

ClinicalTrials.Gov Identifier	Trial Status	Cancer Type	Primary Endpoint
NCT04155359	Recruiting	Bladder Cancer	The test measures up to 280 sncRNA present in urine exosomes and produces a dichotomized assessment of “−1” (no cancer) and “+1” (cancer) based on the expression profiles of the exosomal sncRNAs
NCT05270174	Not yet recruiting	Preoperative Diagnosis of Lymphatic Metastasis in Patients with Bladder Cancer	Explore Whether lncRNA-ElNAT1 in Urine Exosomes Can be Used as a New Target for Preoperative Diagnosis of Lymph Node Metastasis

## Data Availability

Not applicable.

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
