# Peer review of "Applications of Exosomes in Diagnosing Muscle Invasive Bladder Cancer"

_pharmaceutics, 2022, doi:10.3390/pharmaceutics14102027_

Round 1
Reviewer 1 Report
The review article "Applications of Exosomes in Diagnosing Muscle Invasive Bladder Cancer " by walker et al is a nice effort to summarize the recent study about the exosomes in Muscle Invasive Bladder Cancer (MIBC). In this study authors have summarized the recent study related to EV and its diagnostic application in MIBC. Indeed EV has a potential to revolutionize the diagnostic world due to various reasons. This manuscript is very well written and easy to understand and reads well throughout. Authors have nicely put together the important discoveries related to urinary biomarkers conducted in recent years. I believe this manuscript will definitely benefit the scientist working in this field. I would advise authors to improve the manuscript by adding two sections. 1. How the EV is involved in the MISC progression. This section can be added before section two. 2. The technical limitation or challenges to use EV as biomarker. This section could include the technology to isolated urinary exosomes and their characterization.
Author Response
We appreciate reviewer’s recognition of our review manuscript in the field of exosome diagnosis for MIBC. Following reviewer’s suggestions, we have included one paragraph in the page 5 for discussing the Exosome involvement in the MIBC progression. We also added one paragraph in the page 9 to discuss the technical limitations or challenges to use EV as biomarkers, including the isolation technology and characterization limitations.
Reviewer 2 Report
Dear Editor,
The review of Walker et al is comprehensive of many studies on urine exosomes and their use to monitor and manage bladder carcinoma
However, some sentences need to be modified and a clinical trials Table as follows.
please change the "liquid tissues" sentence in lane 97 with more appropriate words (urine is not a tissue)
please add that exosome markers found urine exosome are common to different tumor types one example is Edil3 is expressed also in sarcoma exosomes modulation angiogenesis doi: 10.1038/s41419-021-04069-
Could the author add a Table of clinical trials ongoing and also closed that already use or test exosomes as markers
Could the author also add a paragraph including the technique for exosome preparations approved by FDA and extracellular vesicle society
Author Response
We thank for reviewer insightful suggestions. We have followed suggestions to make modifications as below:
- The line 97 “liquid tissues” has been changed to “samples”
- We also added the correct reference for exosome biomarker EDIL3 and revised as “It should be noted that several of these biomarkers are found in other tumor types and therefore not unique to bladder cancer, such as EDIL-3 found also in sarcomas exosomes modulating angiogenesis (95).”
- We also added a table to highlight and summarize the exosome-based diagnostic clinical trials for bladder cancer in the page 8 as the table 3. For all cancer types, there are nearly 100 clinical trials using exosomes as the biomarkers. We reviewed and discussed current status by adding one paragraph in the page 8.
- We also added one paragraph in the page 9 to review and discuss the techniques used for exosome isolation and characterization accepted by ISEV society for complying with MISEV guidelines.